# Incorporation of 5-Bromo-2′-deoxyuridine into DNA and Proliferative Behavior of Cerebellar Neuroblasts: All That Glitters Is Not Gold

**DOI:** 10.3390/cells10061453

**Published:** 2021-06-10

**Authors:** Joaquín Martí-Clúa

**Affiliations:** Unidad de Citología e Histología, Departament de Biologia Cellular, de Fisiologia i d’Immunologia, Facultad de Biociencias, Institut de Neurociències, Universidad Autónoma de Barcelona, Bellaterra, 08193 Barcelona, Spain; Joaquim.marti.clua@uab.es; Tel.: +34-93-581-1666

**Keywords:** prenatal life, perinatal life, 5-bromo-2′-deoxyuridine, cell cycle, cerebellar neuroepithelium, external granular layer, neurogenetic timetables, neurogenetic gradients, apoptosis

## Abstract

The synthetic halogenated pyrimidine analog, 5-bromo-2′-deoxyuridine (BrdU), is a marker of DNA synthesis. This exogenous nucleoside has generated important insights into the cellular mechanisms of the central nervous system development in a variety of animals including insects, birds, and mammals. Despite this, the detrimental effects of the incorporation of BrdU into DNA on proliferation and viability of different types of cells has been frequently neglected. This review will summarize and present the effects of a pulse of BrdU, at doses ranging from 25 to 300 µg/g, or repeated injections. The latter, following the method of the progressively delayed labeling comprehensive procedure. The prenatal and perinatal development of the cerebellum are studied. These current data have implications for the interpretation of the results obtained by this marker as an index of the generation, migration, and settled pattern of neurons in the developing central nervous system. Caution should be exercised when interpreting the results obtained using BrdU. This is particularly important when high or repeated doses of this agent are injected. I hope that this review sheds light on the effects of this toxic maker. It may be used as a reference for toxicologists and neurobiologists given the broad use of 5-bromo-2′-deoxyuridine to label dividing cells.

## 1. Introduction

The thymidine analogue 5-bromo-2′-deoxyuridine (BrdU) is a pyrimidine 2′-deoxyribonucleoside compound having 5-bromouracil as the nucleobase. This agent is permanently incorporated into the DNA during the synthetic phase of the cell cycle. It has been argued that gene duplication, DNA repair or apoptotic cellular events might contribute to BrdU labeling in vivo. Therefore, this marker is considered as an indicator of DNA synthesis and not the capacity to divide [1,2,3].

BrdU has provided new opportunities for analyzing the proliferative behavior of neuroblasts and to infer the generation of neurons during the development of the central nervous system at the baseline level and under several experimental conditions that disturb the basal status [4,5,6]. Currently, BrdU labeling is the most widely used procedure for studying cell-cycle phases and durations as well as to identify dividing neuroblasts and follow their fate. The random incorporation of this agent into the DNA disturbs its composition and sequence and, therefore, BrdU presents in a myriad of negative effects because its toxicity affects the number and distribution of tagged cells [3,7,8,9,10]. The detrimental effects of BrdU administration are present from prenatal life to adulthood. In this context, when the adult neurogenesis is considered, the possibility of false labeling or the incorrect interpretation of such labeling have indicated that the existence of functional neurogenesis in the adult human brain is questionable. For example, it has been reported that in postmortem brain samples from cancer patients injected with BrdU, some BrdU-stained cells were co-labeled with NeuN in the ventricular–subventricular zone and the dentate gyrus of the hippocampal formation. It was reported in the same study that the amount of BrdU-reactive cells decreased with the longest interval between BrdU administration and histological evaluation. From these results, it was proposed that, in humans, continued neurogenesis exists. However, these data can be interpreted in a different way, e.g., a process not associated with neuroblasts proliferation. Damaged neurons, attempting to repair themselves or due to the activation of apoptotic events, incorporate BrdU and continue to die so that eventually the longer the waiting period before histological analysis, less BrdU-positive cells there are [11]. On the other hand, a way to study, in humans, the process of generating neurons from adult neural precursors is to detect, in the same tissue section, specific antigens for cell proliferation and doublecortin-reactive cells. The controversy is generated when BrdU-immunoreactive cells can be explained by processes not associated with cell division (DNA repair or apoptotic events) and doublecortin-stained cells do not present substantial proliferative activity. These results can be interpreted as the existence of young neurons, which maturation, in the human brain, might take years. Thus, a slow and delayed development of young neurons may replace neurogenetic processes in the adulthood [11,12]. Despite these data, BrdU has been considered that it is relatively benign and the consequences on the progression of the cell-cycle, migration, and fate are usually negligible or understated.

In light of the above, I began a set of experiments to determine the effects of BrdU exposure during the prenatal and perinatal development of the cerebellum. I show here the results of my research indicating that the effect of the incorporation of BrdU into newly synthesized DNA may lead to inaccurate results. Three aspects were addressed. In the first of these, it was studied whether a single injection of BrdU at doses ranging from 25 to 300 µg/g, modifies the development of the cerebellar neuroepithelium. In the second, it was compared the effects of a single-dose of BrdU, at doses ranging from 50 to 300 µg/g, on cell cycle parameters and phase durations in the cerebellar external granular layer neuroblasts. In the third procedure, to know whether there are any differences between BrdU or tritiated thymidine ([^3^H]TdR) labeling, it was determined whether the administration of several doses of both markers, by a progressively delayed cumulative labeling method in utero, modify the developmental timetables and neurogenetic gradients of PCs and DCN neurons.

## 2. The Thymidine Analogue 5-Bromo-2′-Deoxyuridine: An Overview

5-iodo-2′deoxyuridine, 5-chloro-2′deoxyuridine, 5-ethynyl-2′-deoxyuridine, and BrdU are halogenated thymidine analogs [7]. The latter is a chemically synthesized bromine-labeled base analogue that competes with thymidine for incorporation into DNA. Once integrated into the new DNA, BrdU will remain in place and be passed down to daughter cells following mitosis [8,13]. Since the introduction of monoclonal antibodies against BrdU [14], an increasing number of immunostaining assays have been described for the detection of BrdU-labeled cells [10,15]. This nucleoside has provided new advances for investigating several issues during the development of the central nervous system, including developmental timetables, neurogenetic gradients, cell cycle parameters, and cell lineage in a variety of animals such as insects [16], birds [17], and mammals [18,19,20].

BrdU is generally thought to be a relatively benign substitute for the endogenous thymidine. The effects of the incorporation this thymidine analogue on cell proliferation, migration, and differentiation are frequently neglected. BrdU can cause unforeseen problems [21]. In this context, it has been revealed that this agent is an anticancer drug, and in combination with secondary stressors, such as ionizing radiation, BrdU presents adverse consequences for cancer cells [22]. In line with this scenario, previous reports have denoted negative effects of BrdU incorporation on the proliferative dynamics of mouse and human fibroblasts, and adult neural progenitor cells in vitro [8,23,24,25].

Several studies have indicated that BrdU produces sister-chromatid exchanges and double-strand breaks [26]. Other authors have shown that the exposure of rodent neural stem cells to BrdU gives rise to a decrease in the methylation of the DNA [27]. It has also been demonstrated that this synthetic halogenated pyrimidine induces alterations during the growth of the chick dorsal telencephalon [28]. In mammalian embryos, BrdU is toxic to cultured embryos [29], and has a harmful effect on the embryonic development of the neocortex, the striatum [30,31], the cerebral cortex [3], and the cerebellum [9,10]. Deleterious effects on neuron fate [3,7], as well as senescence in several cell types [32,33] including neuronal progenitors [24] have also been reported. These results suggest that BrdU may not be as safe as we thought. Despite that, the effects of the incorporation of this genotoxic agent into DNA on cell proliferation, migration and differentiation are not taken into account. This is particularly important when high doses of BrdU are injected.

## 3. The Cerebellum: A Model to Assess the Effects of Bromodeoxyuridine Administration

The cerebellum is an ancient and a prominent part of the vertebrate central nervous system. It presents a great variability in size and complexity from cyclostomes to humans [34,35,36]. In mammalians, the cerebellum presents a central vermis and hemispheres laterally situated. The first of these has a characteristic pattern of folia separated by fissures running perpendicular to the antero–posterior axis that is distinct from the hemispheres [37,38]. The cerebellum shows a large cortical component, the cerebellar cortex, and a set of nuclei located close to the roof of the fourth ventricle deep within the white matter or cerebellar medulla. They are called deep cerebellar nuclei or deep nuclei [39,40].

In adults, the cerebellar cortex shows a regular trilaminate architecture comprising a ganglionar layer sandwiched between an outer molecular layer and an inner granular layer. The neuronal component of these layers is integrated in a nucleocortical network [41,42], and presents distinctive dendritic arborizations, axons, and perikaryal as well as molecular markers [43,44,45,46]. This regular cytoarchitecture has allowed us to ascertain the successive phases in the ontogeny of the cerebellum, and to analyze the cellular and molecular mechanisms involved in its development [39,47].

Previous neuroembryological studies have shown that the cerebellum arises from the rhombomere 1, which comprises the most anterior part of the hindbrain [37,48]. Expression of the repressive homeobox genes *orthodenticle homolog 2* (Otx2) and *gastrulation brain homeobox 2* (Gbx2) establish the midbrain–hindbrain boundary and form the isthmic organizer [49,50,51]. It has been observed that the isthmus, the neural tissue located at the midbrain–hindbrain junction, plays an important role in the establishment of the cerebellar territory [52]. The organizing activity of the isthmic tissue is mediated by the expression of the *fibroblast growth factor 8* gene, which encode the fibroblast growth factor 8, a protein required for orchestrating multiples stages of cerebellar development [51]. The expression of the *fibroblast growth factor 8* gene is regulated by several genes, which are involved in the control of the cerebellum development. Among these genes are the engrailed 1 and 2 [53], and the Pax 2 and Pax 5 [37,54].

The establishment of the cerebellar territory is followed by the formation of two primary germinative areas: the cerebellar neuroepithelium, defined by Ptf1a-expression progenitors, and the rhombic lip, defined by progenitor cells expressing Atoh1 [55]. Many lines of evidence have indicated that GABAergic neurons such as Purkinje cells (PCs), inhibitory interneurons and some deep cerebellar nuclei (DCN) neurons originate from the cerebellar neuroepithelium. By contrast, glutamatergic neurons including DCN neurons, unipolar brush cells and the projection neurons of the precerebellar nuclei arise from progenitors located in the rhombic lip [37,47]. The granule cells are also glutamatergic. Their neuroblasts precursors also arise from the rhombic lip, but they migrate over the surface of the cerebellar anlage to form a temporary matrix, the external granule layer (EGL) [45,56]. These data have revealed that cerebellar neurogenesis is a process strictly compartmentalized.

The cerebellum has been selected as a model to study and interpret the effects of BrdU because this encephalic region is highly vulnerable to intoxication and poisoning. It is the main target of several environmental toxins including mercury, lead, manganese, and toluene/benzene derivates, and drugs such as anticonvulsants, lithium salts, and antineoplastics [57]. The analysis of the cerebellar development can serve as a model to know the effects of BrdU on the development of other areas of the central nervous system.

## 4. Effect of Bromodeoxyuridine Exposure on the Cerebellar Neuroepithelium

During the early development of the cerebellum, neuroblasts located in a specialized germinal matrix, the neuroepithelium, gives rise to several types of neurons including interneurons, PCs and some DCN neurons [47]. Previous research has revealed that that the dose of BrdU usually used in many laboratories (50 µg/g) is well tolerated, when administered in the embryonic period, and produces no cytotoxic effects [58,59]. These observations deserve attention because other data from the literature have been reported that the administration of BrdU affects the cell cycle progression of neural stem cells [60]. In addition to that, when injected in the prenatal life, low doses of BrdU compromise the number and distribution of spatial distribution of cells in the cerebral cortex of monkeys [3]. Moreover, there are also proofs revealing that the administration of this marker in utero alters neuroblast proliferation, migration, and patterning of the cerebellum [9,10]. These results suggest that BrdU may not be as safe as we thought. In this section, evidences are presented indicating that high doses of BrdU interferes with cell proliferation and promotes apoptotic cellular events in the neuroepithelium.

Pregnant rats were injected at embryonic day (E) 13 with a single dose of saline or BrdU at doses ranging from 25 to 300 µg/g. Embryos were removed by caesarian at regular intervals from 5 to 30 h after agent exposure. To determine the effect of BrdU administration on cerebellar neuroepithelial cells several parameters were quantified: (I) density of mitotic figures, (II) BrdU, and (III) proliferating cell nuclear antigen (PCNA)-positive cells. Results indicate that, irrespective of the analyzed survival times, the density of the afore-mentioned parameters was close in animals exposed to doses of BrdU ranging from 25 to 75 µg/g (Figure 1). Interestingly, no statistical differences were observed in comparison with rats administered with saline. However, when doses of 100 to 300 µg/g of BrdU were studied, it was observed the analyzed parameters were decreased in comparison to the saline group. The dose of 300 µg/g of BrdU produced the most detrimental effects (Figure 1). These results have indicated that a single low dose of BrdU (25 to 75 µg/g) does not alter the proliferative behavior of the neuroepitelial neuroblasts. Despite this, a more protracted effect on neuroepithelial neuroblasts cannot be excluded, i.e., cell differentiation and final fate.

On the other hand, to discover whether BrdU-injection leads to apoptotic degeneration, the density of terminal deoxynucleotidyl transferase dUTP nick-end labeling (TUNEL) and active caspase-3 were quantified. Transmission electron microscopy was carried out to confirm apoptotic cell death. Results showed that, in comparison to saline injected rats and irrespective of the analyzed survival times, the density of TUNEL-positive and caspase-3 reactive neuroblasts were similar in animals exposed to doses of BrdU ranging from 25 to 75 µg/g of BrdU (Figure 2). No statistical differences were seen in comparison to rats administered with saline. When doses of 100 to 300 µg/g of BrdU were considered, the density of these parameters increased, indicating that high doses of this brominated thymidine led to the activation of apoptotic cellular events (Figure 2).

In order to verify the apoptotic state of neuroepithelial cells, neuroblasts treated with saline of BrdU (100 to 300 µg/g) were examined with the transmission electron microscopy (TEM). The earliest ultrastructural alterations were the presence of nuclear chromatin clumps of high electron density in close contact with the inner nuclear envelope. Another important feature was the presence of clusters of approximately round electron-dense apoptotic bodies containing very condensed dark chromatin masses. In some cases, the apoptotic bodies were broken, and their content extruded into the cytoplasm (Figure 3).

At least, two observations follow from these data. (I) A single low dose of BrdU (25 to 75 µg/g) alters neither the cell-cycle progression nor promotes apoptosis, and (II) high doses of BrdU (100 to 300 µg/g) during the early prenatal life activates apoptotic cellular events, leading to an important depletion of neuroblasts.

## 5. Effects of Bromodeoxyuridine Exposure on the Cerebellar External Granular Layer Neuroblasts

The EGL is a transient proliferative matrix of growing cerebellar folia located beneath the piamater. In most of the vertebrates, this structure increases in thickness during an initial period of time as a result of proliferative activity of its neuroblasts [45,56]. Previous reports have revealed that BrdU is a radiosensitizer [22]. However, its therapeutic potential as an antitumoral agent is independent of secondary stressors. This is because BrdU inhibits cancer cell proliferation both in vitro and in vivo [22,61,62]. It has been reported that this agent alters cell cycle dynamics in neurosphere cultures derived from adult rat brain [8]. In this section, evidences are provided revealing that, at postnatal day (P) 9, a single high dose of BrdU interferes with the cell cycle progression of the EGL neuroblasts.

Rats were injected with BrdU at doses ranging from 50 to 300 µg/g. This thymidine analogue was supplied according to two schedules: in the first of these, rats were allowed to survive for 0.5, 1, 1.5, and 2 h after BrdU administration. In the second, animals were sacrificed 1 h later, and at regular intervals from 2 h to 26 h (spaced at two-hour intervals) following BrdU administration. The labeling index (LI) was determined as a percentage of BrdU-stained interphase nuclei per total number of scored EGL neuroblasts. Mitotic figures were examined for the presence or absence of 3,3′-diaminobenzidine-H_2_O_2_ reaction product in BrdU immunostained sections. The duration and phases of the cell cycle were inferred from the graphic representation of the percentage of BrdU-reactive mitoses plotted as a function of survival time after agent injection [15,20,63].

Figure 4 summarizes, in rats injected with BrdU at doses ranging from 50 to 300 µg/g, the variation in the frequency of labeled neuroblasts after different survival times (from 0.5 to 2 h). Data analysis indicated that, from 0.5 to 2 h after marker injection, the LI was highest using 50 µg/g of BrdU. It was followed by 100 and then by 200 µg/g. Does of 300 µg/g provided the lowest values.

Incorporation of BrdU into proliferating neuroblasts labels a cohort of asynchronous cycling cells in S-phase. Phase durations and cell-cycle length of the EGL neuroblasts can be quantitatively examined from the rhythmic appearance and disappearance of labeled mitotic cells. The variation that took place in the fraction of labeled mitoses from 1 to 26 h after BrdU exposure is indicated in Figure 5A. Cell kinetic parameters were inferred from Figure 4 and Figure 5A. The time required for completion of each phase of the cell cycle is displayed in Figure 5B. Current results denoted that different doses of BrdU affect the parameters used for estimated the time required to complete the entire cell cycle.

One fact emerges from these experiments. A single dose of 50 µg/g of BrdU has no apparent harmful effects on the cell cycle progression, which suggest that this dose is appropriate, and it provides accurate results. Higher doses (100 to 300 µg/g) altered the detection of BrdU-immunoreactive cells. As the duration and phases of the cell cycle are inferred in accordance with BrdU detection, an effect on this detection can render the measurement of the cell cycle inaccurate.

## 6. Inferring Purkinje Cells and Deep Cerebellar Neurons Developmental Timetables with Bromodeoxyuridine or Tritiated Thymidine. What Is the Most Suitable Marker?

BrdU and tritiated thymidine ([^3^H]TdR) are two markers of DNA synthesis. They have produced important insights into the neural mechanisms of the central nervous system development including cell kinetics, neuron production and migration [15,18,19,39,64,65]. Both nucleosides have advantages and disadvantages. For example, both BrdU and [^3^H]TdR produce cytotoxic and teratologic effects when high doses of these are supplied [9,13,30]. In addition, the intensity of [^3^H]TdR labeling is stoichiometric while BrdU is not [66,67,68]. BrdU can rapidly be revealed by immunohistochemical procedures, whilst [^3^H]TdR autoradiography is more expensive, requires a special laboratory and it takes a lot of days to develop a picture.

Previous reports have indicated that, in the cerebral cortex of macaque monkeys exposed to either BrdU or [^3^H]TdR as embryos, quantitative differences in the number and placement of tagged neurons exist [3]. In this section, proofs are presented indicating that repeated injections of BrdU produce, in comparation with several [^3^H]TdR administrations, systematic differences in the pattern of PCs and deep neurons neurogenesis as well as in spatial location of these macroneurons.

To study the generation and spatial location of rodents PCs and DCN neurons, both markers were administered following a progressively delayed cumulative labeling method [9,64]. This method consists of injecting pregnant dams with BrdU or [^3^H]TdR in an overlapping series in accordance with the following time-windows: embryonic day (E)11–12, E12–13, E13–14, and E14–15. Six intraperitoneal injections of 6 mg BrdU (10 mg/mL in sterile saline with 0.007 N sodium hydroxide) 8 h apart were supplied [9]. Whereas, [^3^H]TdR (5 µCi/g of body weight) was subcutaneously supplied. Two doses were administered on consecutive days [4,69]. PCs were analyzed in four compartments of the cerebellar cortex (vermis, paravermis, and medial and lateral hemispheres). DCN neurons were studied in the fastigial, interpositus, and dentate nuclei. Animals were sacrificed in adulthood.

Variations in the percentage of tagged neurons after different labeling times with BrdU or [^3^H]TdR are shown in Figure 6 and Figure 7 for PCs and DCN neurons, respectively. It is observed that, in each time-window and in each cerebellar compartment, the proportion of [^3^H]TdR labeled PCs is higher that the percentage of BrdU tagged PCs. Similar results were obtained when DCN neurons were taken into account.

In a subsequent study, the neurogenetic timetables of PCs and DCN neurons were built. In Figure 6 (for PCs) and Figure 7 (for DCN neurons), the percentages of produced neurons are plotted against the time. My results have revealed that, with the exception of medial hemispheres at E11, more BrdU than [^3^H]TdR labeled PCs are produced on embryonic day 10 and 11 in the remain studied areas. The same takes place in each deep cerebellar nucleus studied at E10. However, when frequencies of newborn macroneurons were inferred from E12 to E13, more PCs were produced when the radioactive precursor was used. Similar results were seen on E12 when all the deep nuclei were examined.

The cerebellum develops following a well-known spatiotemporal sequence of events. There is evidence indicating that PCs and DCN neurons are distributed throughout the cerebellum according to precise neurogenetic gradients [37]. By examining the above-inferred times of neuron origin, it is possible to classify PCs and DCN neurons among those produced from E10 to E11 (early-born) and those generated from E12 to E14 (late-generated). In Figure 8 (for PCs) and Figure 9 (for DCN neurons), the proportions of each population fractions are compared for BrdU and [^3^H]TdR. My results have shown that, in the analyzed cerebellar compartments as well as in each deep nucleus, proportions of early-produced BrdU-reactive cells were always higher than percentages of neurons tagged with [^3^H]TdR. The opposite pattern was found when late-produced neurons were studied, revealing that significant differences in the settling pattern of PC and DCN neurons exit depending on whether BrdU and [^3^H]TdR are used.

It can be drawn as a general conclusion that, in comparison to several doses of [^3^H]TdR, to infer neurogenetic timetables and neurogenetic gradients using several doses of BrdU is less accurate. This is presumably due to the BrdU toxicity. The incorporation of this marker into DNA produces base pairing of the bromouracil with guanine instead of adenine. This substitution is involved in mutations and breaks in double-stranded DNA leading to detrimental effects on the differentiation and survival of neuroblasts [8] and on neuron fate and function [3,26,28]. Interestingly, I have found no damage to cell-cycle progression after [^3^H]TdR exposure. This was expected because the difference between [^3^H]TdR and the thymine is only an extra neutron in a hydrogen atom. Therefore, as suggested by Duque and Rakic [3,7], DNA with [^3^H]TdR possibly reflects closer DNA in the non-injected animal.

## 7. Conclusions

The synthetic halogenated pyrimidine analogue BrdU is a marker of DNA synthesis. BrdU has provided important clues about the development of the central nervous system under different experimental contexts. Despite the fact that BrdU toxicity has been demonstrated, it is generally thought that this agent is a benign substitute for the [^3^H]TdR. The effect of the incorporation of BrdU on proliferating neuroblasts is frequently neglected by investigators. This current review has implications for the interpretation of the results obtained by this marker to investigate the cell cycle of neuron precursors as well as the genesis of neurons. This is particularly important when a single high dose of BrdU is used because effects on proliferative behavior and the activation of apoptotic cellular events may lead to false results in the identification of proliferative neuroblasts. The same occurs when repeated injections of BrdU are supplied following a progressively delayed cumulative labeling method. Thus, data obtained with this thymidine analogue should be interpreted with caution. It is proposed that to label proliferating neural progenitors, a single pulse of BrdU (until 50 µg/g) has no apparent harmful effects. I hope that this current review may be also useful when studying and interpreting cell birth with BrdU labeling in the rest of the central nervous system.

## Figures and Tables

**Figure 1 cells-10-01453-f001:**
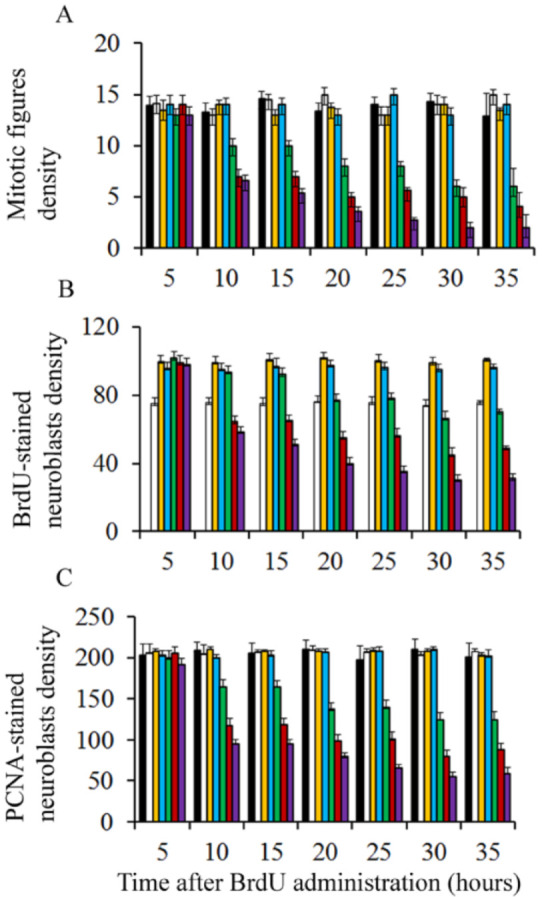
Mean values for mitotic figure (**A**), BrdU-reactive neuroblasts (**B**), and PCNA-stained cells density (**C**) in the cerebellar neuroepithelium of rodents injected with saline (black columns), 25 µg/g (white columns), 50 µg/g (yellow columns), 75 µg/g (blue columns), 100 µg/g (green columns), 200 µg/g (red columns), and 300 µg/g of BrdU (purple columns).

**Figure 2 cells-10-01453-f002:**
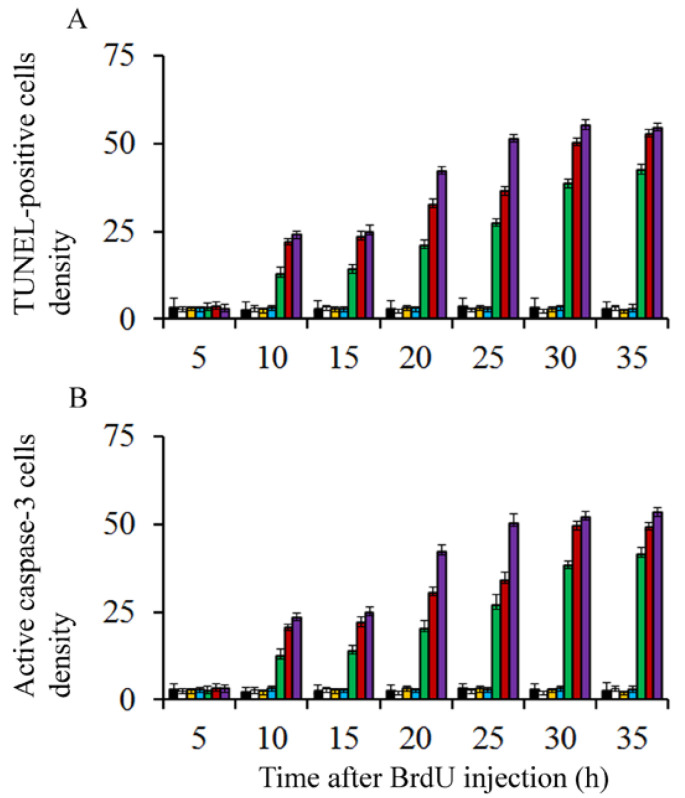
Mean values for TUNEL-stained (**A**) and active caspase 3-reactive neuroblasts (**B**) in the cerebellar neuroepithelium of rodents injected with saline (black columns), 25 µg/g (white columns), 50 µg/g (yellow columns), 75 µg/g (blue columns), 100 µg/g (green columns), 200 µg/g (red columns), and 300 µg/g of BrdU (purple columns).

**Figure 3 cells-10-01453-f003:**
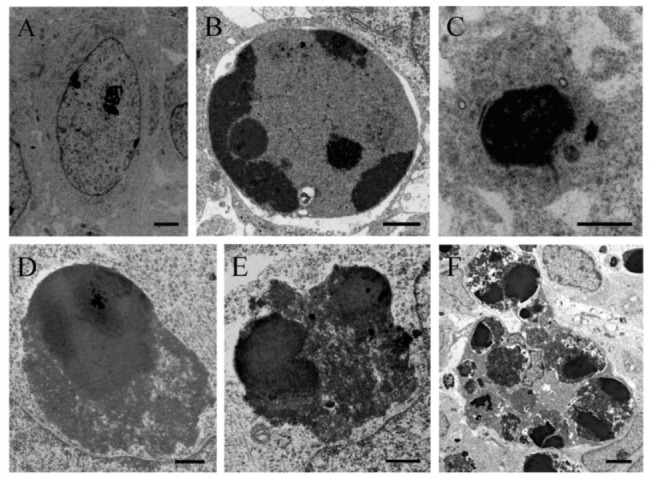
Electron micrograph of a healthy cell (**A**) and apoptotic cellular profile (**B**–**F**) from the neuroepithelium 10 h following BrdU treatment (100 µg/g). (**C**) Ultrastructural morphology of an apoptotic cell. The masses of compact chromatin display a high electron density and a homogeneous texture. They are associated with an intact nuclear envelope. (**D**–**F**) Typical electron-dense apoptotic bodies. (**D**–**E**) Examples of apoptotic bodies releasing their contents into the cytoplasm. Copyright © 2021, Wiley. Adapted with permission from Rodríguez-Vázquez, L.; Martí, J. (2000). Administration of 5-bromo-2′deoxyuridine interferes with neuroblast proliferation and promotes apoptotic cell death in the rat cerebellar neuroepithelium. *J. Comp. Neurol.* **2021,** *529*, 1081–1096. Scale bar: 2 µm (**A**), 1 µm (**B**), 2.5 µm (**C**), 1 µm (**D**), 0.5 µm (**E**), 2 µm (**F**).

**Figure 4 cells-10-01453-f004:**
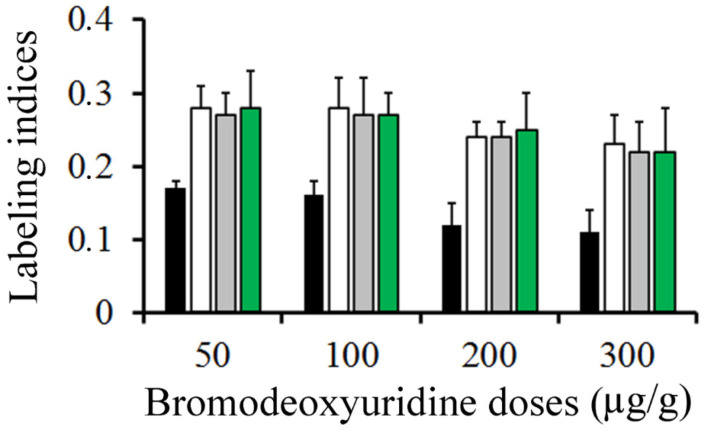
Mean values of labeling indices in the external granular layer. Animals were sacrificed at 0.5 (black columns), 1 (white columns), 1.5 (grey columns), and 2 h (green columns) after a single injection of bromodeoxyuridine at doses ranging from 50 to 300 µg/g.

**Figure 5 cells-10-01453-f005:**
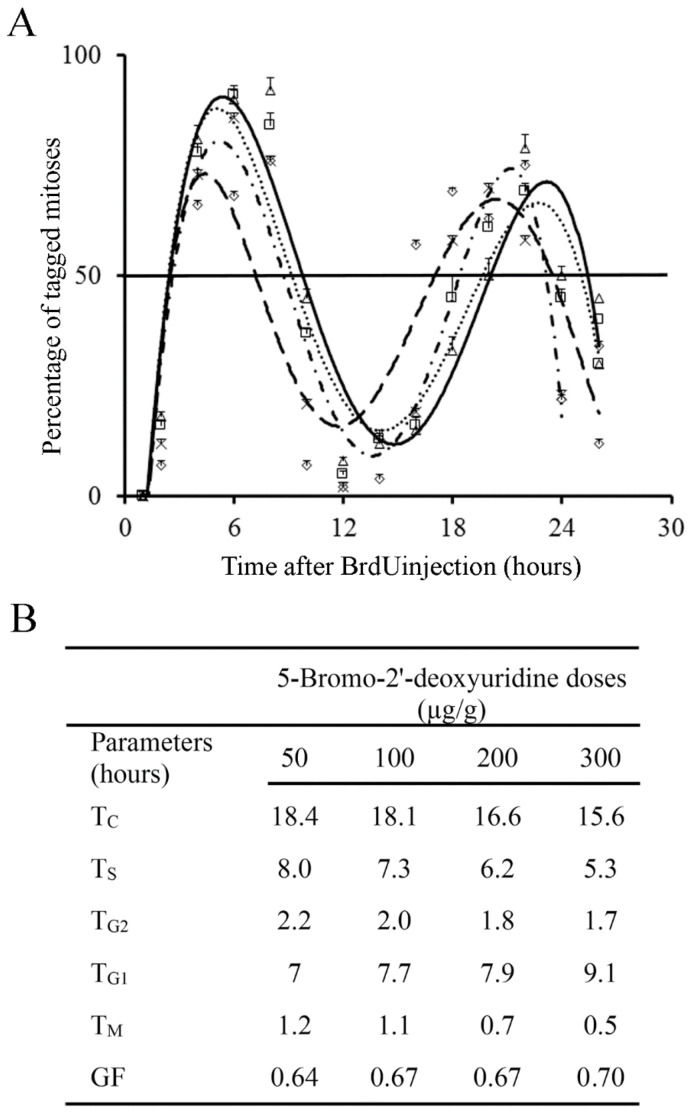
(**A**) Frequency of labeled mitoses in the external granular layer at successive times following a single injection of 50 µg/g (open triangles and solid black lines), 100 µg/g (open squares and dotted line), 200 µg/g (crosses and short broken lines), and 300 µg/g of BrdU (diamonds and large broken lines). Data are expressed as mean ± SEM. (**B**) Values for phase durations and cell-cycle length of the EGL cells. T_C_: duration of the whole cell cycle. T_S_: duration of the S-phase. T_G2_: duration of the G2 phase. T_G1_: duration of the G1 phase. T_M_: duration of the mitotic phase. GF: growth fraction.

**Figure 6 cells-10-01453-f006:**
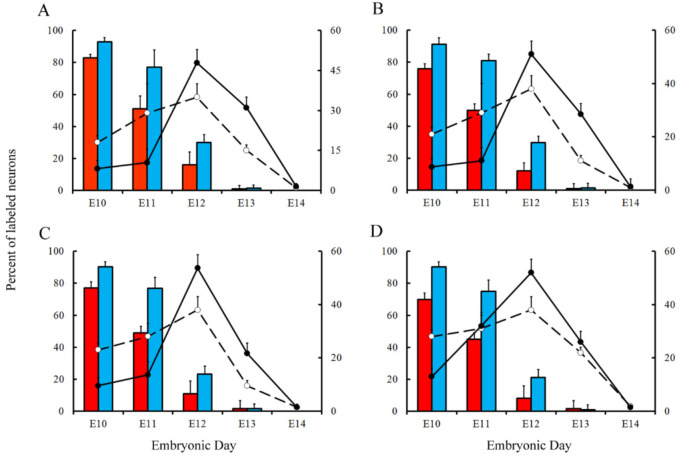
Comparison of Purkinje cells neurogenetic patterns in animals injected with bromodeoxyuridine (red columns) or [^3^H]TdR (blue columns) at the level of the vermis (**A**), paravermis (**B**), medial (**C**), and lateral hemispheres (**D**). Developmental timetables of Purkinje cells were inferred using bromodeoxyuridine (open circles and dashed lines) or [^3^H]TdR (closed circles and solid black lines). E: embryonic day. Values are expressed as mean ± SEM.

**Figure 7 cells-10-01453-f007:**
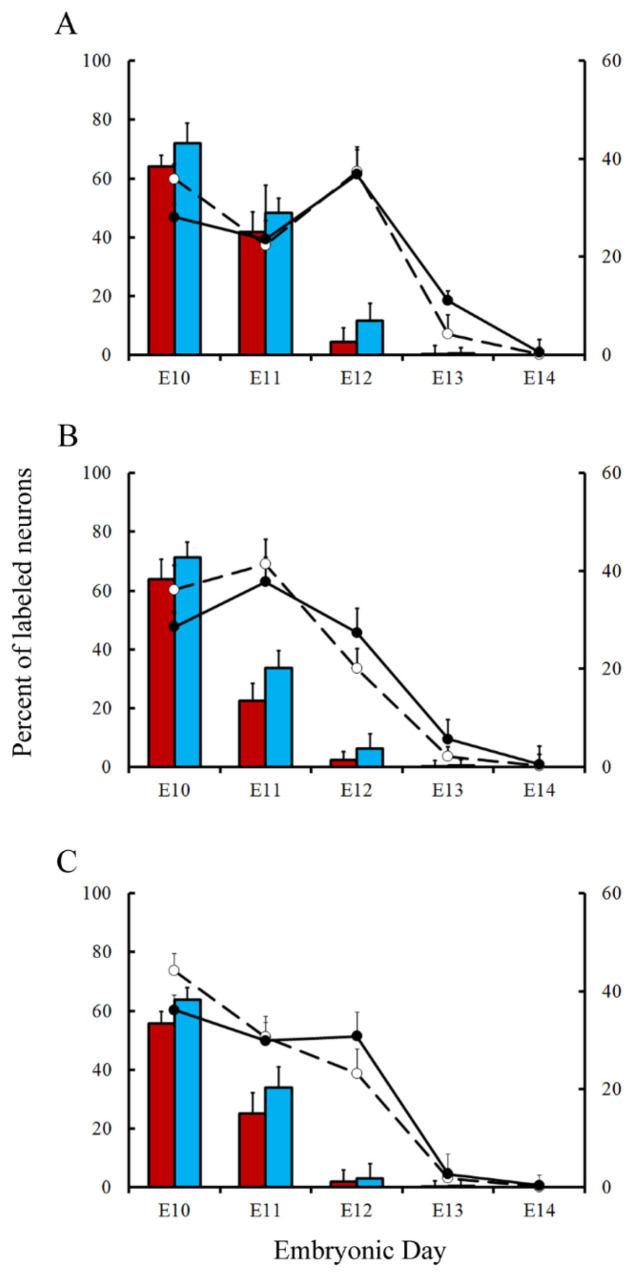
Comparison of deep nuclei neurons neurogenetic patterns in animals injected with bromodeoxyuridine (red columns) or [^3^H]TdR (blue columns) at the level of the fastigial (**A**), interpositus (**B**), and dentate (**C**) nuclei. Developmental timetables of deep nuclei neurons were inferred using bromodeoxyuridine (open circles and dashed lines) or [^3^H]TdR (closed circles and solid black lines). E: embryonic day. Values are expressed as mean ± SEM.

**Figure 8 cells-10-01453-f008:**
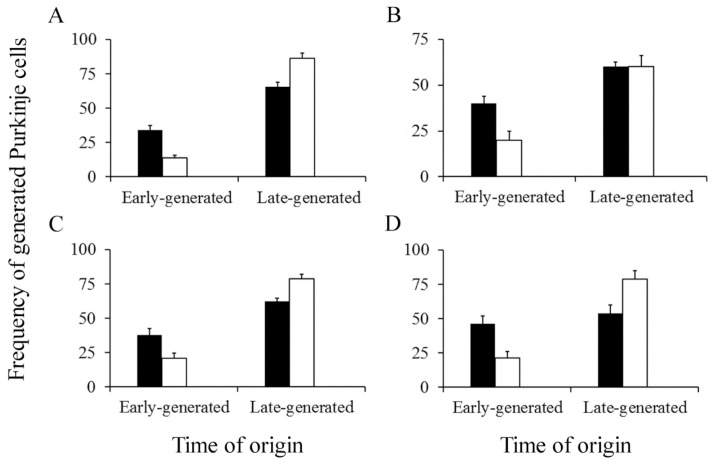
Percentage of early and late-produced Purkinje cells scored at the level of the vermis (**A**), paravermis (**B**), medial (**C**), and lateral hemispheres (**D**). Black columns indicate animals injected with bromodeoxyuridine and white columns animals administered with [^3^H]TdR. Values are expressed as mean ± SEM.

**Figure 9 cells-10-01453-f009:**
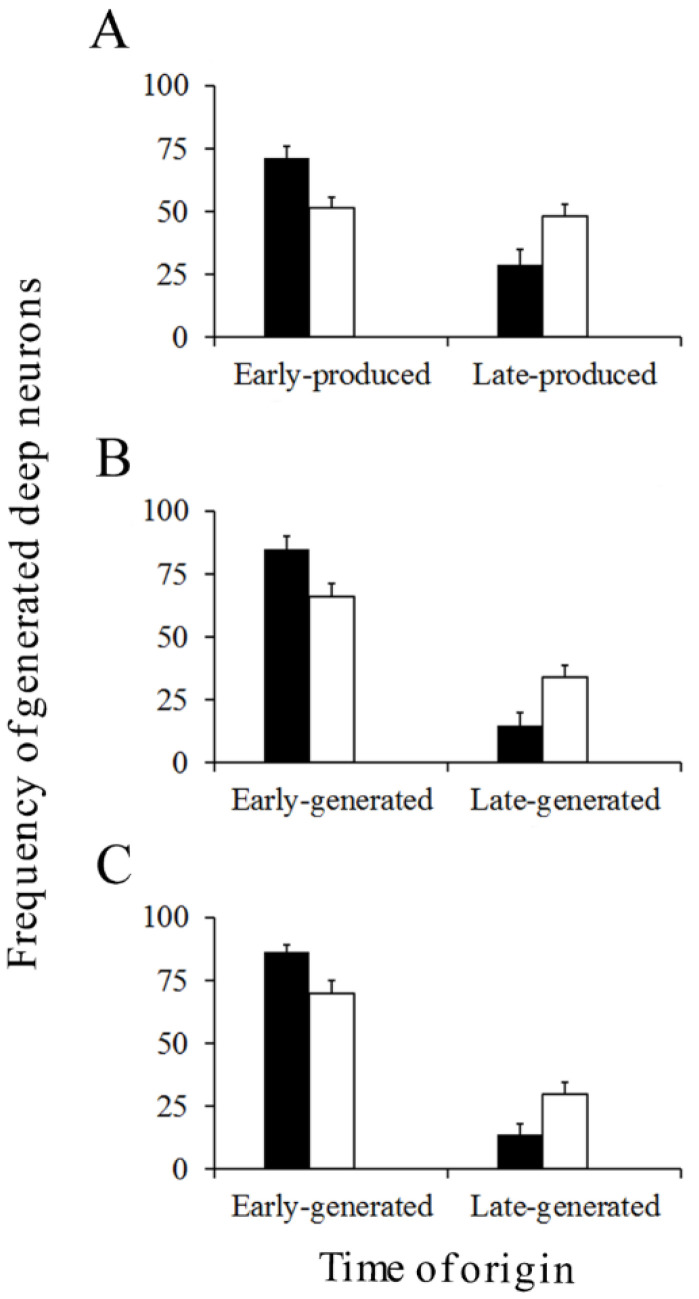
Percentage of early and late-generated deep nuclei neurons scored at the level of the fastigial (**A**), interpositus (**B**), and dentate (**C**) nuclei. Black columns indicate animals injected with bromodeoxyuridine and white columns animals administered with [^3^H]TdR. Values are expressed as mean ± SEM.

## Data Availability

There are no special databases associated with this manuscript.

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
