# Peer review of "Incorporation of 5-Bromo-2′-deoxyuridine into DNA and Proliferative Behavior of Cerebellar Neuroblasts: All That Glitters Is Not Gold"

_cells, 2021, doi:10.3390/cells10061453_

Round 1

Reviewer 1 Report

Marti-Clua  J Cells (Review)

This is an important, timely and critical review about the values as well as limitations of the use of thymidine analogue 5-bromo-2-deoxyuridine (BrdU) as a marker for cell division and indicator generator of the introduction of new neurons into the adult brain. It is well established that BrdU incorporation indicates DNA replication, which can occur not only during cell division but also during many other conditions, including DNA repair and replication in response to excessive function, hypoxia, cell attempt of repair damages, as well as programmed and induced cell death (apoptosis).  The present review is well composed and written, and it also contains some original data on cell geneses during development of the cerebellum in rat. The paper could be acceptable as is, but I have some minor suggestions to make it more detailed. For example, references are well selected, but it would be useful and informative to cite the recent review on the same topic by Duque and Spector (Brain Structure and Function, 2019). Their review provides details about specific technical problems associated with the use and misuse of this marker as evidence for adult neurogenesis .  

Author Response

REPLY TO REVIEWER 1

Thank you very much for your review of the present manuscript. Your constructive suggestions have been very valuable to us. The following points have been modified in accordance with your recommendations. I think that I have now responded clearly to the reviewer’s comment.

The reviewer wrote (in bold, cursive writing):

This is an important, timely and critical review about the values as well as limitations of the use of thymidine analogue 5-bromo-2-deoxyuridine (BrdU) as a marker for cell division and indicator generator of the introduction of new neurons into the adult brain. It is well established that BrdU incorporation indicates DNA replication, which can occur not only during cell division but also during many other conditions, including DNA repair and replication in response to excessive function, hypoxia, cell attempt of repair damages, as well as programmed and induced cell death (apoptosis).  The present review is well composed and written, and it also contains some original data on cell geneses during development of the cerebellum in rat. The paper could be acceptable as is, but I have some minor suggestions to make it more detailed. For example, references are well selected, but it would be useful and informative to cite the recent review on the same topic by Duque and Spector (Brain Structure and Function, 2019). Their review provides details about specific technical problems associated with the use and misuse of this marker as evidence for adult neurogenesis.

I appreciate this comment and completely agree. In the revised version of this manuscript, the review by Duque and Spector (Brain Struct Function. 2019, 224, 2281-2295) has been added. See pages 2 and 17. Moreover, some examples of improper use of the BrdU are shown.

Special thanks to you for your good comments.

Reviewer 2 Report

In the manuscript entitled: “Incorporation of 5-bromo-2'-deoxyuridine into DNA and prolif-2 erative behavior of cerebellar neuroblasts: all that glitters is not 3 gold”, the Author re-examines and reviews a previously discussed topic concerning the problems/pitfalls encountered when using BrdU as a marker for cell proliferation in the nervous system.

The conclusioni is that low doses can be reliable whereas at higher doses BrdU can give false positive results by marking cells that are not proliferating.  

Though this issue has been discussed in the past, the manuscript is timely, since many Author ignore such problems and several papers continue to describe false-positive data, e,g., in the adult neurogenesis field. This manuscript reminds that several pitfalls can occur when detecting cell division in the nervous system and explains why.

The only limit of the manuscript concerns the choice to restrict the model system to the cerebellum. This is a good model, yet, many pitfalls and controversies arose in the adult neurogenic sites (hippocampus and subventricular zone). At least, the Author might give some examples in which an improper/wrong use of BrdU has been shown to occur.

Finally, since cell proliferation in the nervous system is often detected in association with the marker for immaturity doublecortin (DCX), and since many Authors started to use DCX alone to show the occurrence of neurogenesis, it should be useful to remind that now we know that many populations of DCX-positive neurons can occur in the adult brain in the absence of cell division (the so-called “immature neurons”; La Rosa et al., 2020, Front Neurosci).

Author Response

REPLY TO REVIEWER 2

Thank you very much for your review of the present manuscript. Your constructive suggestions have been very valuable to us. The following points have been modified in accordance with your recommendations. I think that I have responded clearly to the reviewer’s comments.

The reviewer wrote (in bold, cursive writing):

In the manuscript entitled: “Incorporation of 5-bromo-2'-deoxyuridine into DNA and prolif-2 erative behavior of cerebellar neuroblasts: all that glitters is not 3 gold”, the Author re-examines and reviews a previously discussed topic concerning the problems/pitfalls encountered when using BrdU as a marker for cell proliferation in the nervous system.

The conclusion is that low doses can be reliable whereas at higher doses BrdU can give false positive results by marking cells that are not proliferating.  Though this issue has been discussed in the past, the manuscript is timely, since many Author ignore such problems and several papers continue to describe false-positive data, e,g., in the adult neurogenesis field. This manuscript reminds that several pitfalls can occur when detecting cell division in the nervous system and explains why.

The only limit of the manuscript concerns the choice to restrict the model system to the cerebellum. This is a good model, yet, many pitfalls and controversies arose in the adult neurogenic sites (hippocampus and subventricular zone). At least, the Author might give some examples in which an improper/wrong use of BrdU has been shown to occur.

Finally, since cell proliferation in the nervous system is often detected in association with the marker for immaturity doublecortin (DCX), and since many Authors started to use DCX alone to show the occurrence of neurogenesis, it should be useful to remind that now we know that many populations of DCX-positive neurons can occur in the adult brain in the absence of cell division (the so-called “immature neurons”; La Rosa et al., 2020, Front Neurosci).

I appreciate this comment and completely agree. In the revised version of this manuscript, the review by La Rosa et al., 2020 (Front. Neurosci. 2020, 14: 75) has been added. See pages 2 and 17. Moreover, some examples of improper use of the BrdU are indicated.

Once again, thank you very much for your comments and suggests.